# Conversion of Plant Secondary Metabolites upon Fermentation of *Mercurialis perennis* L. Extracts with two Lactobacteria Strains †

**Peter Lorenz [1],\*, Marek Bunse [1], Simon Sauer [1], Jürgen Conrad [2], Florian C. Stintzing [1] and Dietmar R. Kammerer [1]**

[1] Department of Analytical Development & Research, Section Phytochemical Research,
   WALA Heilmittel GmbH, 73087 Bad Boll/Eckwälden, Germany; marek.bunse@wala.de (M.B.);
   simon.sauer@wala.de (S.S.); florian.stintzing@wala.de (F.C.S.); dietmar.kammerer@wala.de (D.R.K.)
[2] Institute of Chemistry, Bioorganic Chemistry, Hohenheim University, Garbenstraße 30, 70599 Stuttgart,
   Germany; juergen.conrad@uni-hohenheim.de
\* Correspondence: peter.lorenz@wala.de; Tel.: +49-7164-930-7099
† Dedicated to Prof. Dr. habil. Dr. h.c. Reinhold Carle on the occasion of his retirement.

**Abstract:** Microbial fermentation of plant extracts with Lactobacteria is an option to obtain microbiologically stable preparations, which may be applied in complementary medicine. We investigated the metabolic conversion of constituents from *Mercurialis perennis* L. extracts, which were prepared for such applications. For this purpose, aqueous extracts were inoculated with two Lactobacteria strains, namely *Pediococcus sp.* (PP1) and *Lactobacillus sp.* (LP1). Both were isolated from a fermented *M. perennis* extract and identified by 16S rRNA sequencing. After 1 day of fermentation, an almost complete conversion of the genuine piperidine-2,6-dione alkaloids hermidine quinone (**3**) and chrysohermidin (**4**)—both of them being oxidation products of hermidin (**1**)—was observed by GC-MS analysis, while novel metabolites such as methylhermidin (**6**) and methylhermidin quinone (**7**) were formed. Surprisingly, a novel compound plicatanin B (bis-(3-methoxy-1*N*-methylmaleimide); **8**) was detected after 6 days, obviously being formed by ring contraction of **4**. An intermediate of a postulated reaction mechanism, isochrysohermidinic acid (**14**), could be detected by LC-MS. Furthermore, an increase in contents of the metabolite mequinol (4-methoxyphenol; **9**) upon fermentation points to a precursor glycoside of **9**, which could be subsequently detected by GC-MS after silylation and identified as methylarbutin (**15**). **15** is described here for *M. perennis* for the first time.

**Keywords:** dog's mercury; lactic acid fermentation; *Pediococcus*; *Lactobacillus*; benzilic acid rearrangement; decarboxylation; de-glycosylation

## 1. Introduction

Numerous biotechnological processes in the food, agriculture and herbal industries are based on lactic acid (LA) fermentation [1–4]. Besides growth inhibition of undesirable or pathogenic bacteria by LA, cell wall degrading enzymes of the lactic acid forming bacteria (LAB) contribute to the maceration of the plant matrix, therefore enhancing the bioavailability of the plant secondary metabolites. In addition, LAB may provide probiotic compounds [4,5]. Consequently, lactofermentation of medicinal plants is a worthwhile process to obtain extracts forgoing the use of organic solvents [6]. In such fermented extracts, genuine plant metabolites are either still intact, or are partially or completely converted into other constituents [7]. This bioconversion of plant constituents is similar to the metabolism found in the human intestine, with its microbiome being composed of a wide range of different LAB [8,9]. Fermentation of herbal extracts may further be used to enhance the functional features of medicinal

plants or diminish toxic side effects [10–13]. We have recently studied the LAB fermentation chemistry of several medicinal plants [14–18]. However, numerous microbial conversion pathways of plant constituents still remain unknown, a reason to put further efforts into this research area.

Dog's Mercury (*Mercurialis perennis* L.) is an old medicinal plant mostly known from ethnomedicine [19]. Nowadays, the herbal parts are mainly used for the preparation of complementary medicinal products, which are externally applied e.g., for the treatment of poorly healing wounds, mammilitis of lactating women, hemorrhoids or conjunctivitis [19]. However, *M. perennis* is also regarded as poisonous when the plant is digested [20,21]. However, novel investigations into the toxicity of *M. perennis* are lacking in scientific literature and reports on fatal (lethal) effects of the herb are ambiguous and were probably due to confusion with annual dog's mercury (*Mercurialis annua* L.) [22]. The latter contains 3-cyanopyridine as a poisonous principle, which is not found in *M. perennis* [23]. We have extensively studied the phytochemistry of this plant genus for the last ten years. Meanwhile, numerous constituents of *M. perennis* and further *Mercurialis* species have been identified, e.g., the piperidine-dione alkaloid hermidin (**1**) [24–26], cinnamic acid depsides, flavonoids [27] and further low-molecular phenolics, terpenoids [26] and *n*-alkyl resorcinols [23,28,29]. Recently, we reported a first study on the metabolism of constituents upon spontaneous fermentation of *M. perennis* extracts, as a result of the action of the wild LAB flora [30]. Even though these investigations revealed the formation of novel metabolites, some of the enzymatic conversion pathways, especially of the N-containing compounds, are still poorly understood.

Consequently, the main focus of the current study was to investigate the conversion of *M. perennis* constituents under the influence of isolated Lactobacteria strains, aiming at the preparation of tailor-made extracts with predefined compound profiles, thus ensuring a constant quality of phytopharmaceutical preparations in the future. Particular attention was paid to the formation of novel metabolites and microbial (LAB) conversion pathways of *M. perennis* constituents, which were studied by GC-MS and LC-MS methods.

## 2. Materials and Methods

### 2.1. Chemicals and Culture Media

MRS agar (*Lactobacillus*-agar according to De Man, Rogosa and Sharpe) was purchased from Merck KGaA (Darmstadt, Germany). MRS broth was obtained from VWR International (Leuven, Belgium). Amberlite® IRA 402 anion exchanger (chloride form) was bought from Sigma-Aldrich Chemie GmbH (Steinheim, Germany), arbutin was from Carl Roth GmbH + Co. KG (Karlsruhe, Germany). *N*-Methylmaleimide was obtained from ABCR GmbH (Karlsruhe, Germany) and dibenzo-18-crown-6 from Merck Schuchardt OHG (Hohenbrunn, Germany). Isochrysohermidin (mixture of *d,l*- and *meso*-dimethyl-2,2',5,5'-tetrahydro-5,5'-dihydroxy-4,4'-dimethoxy-1,1'-dimethyl-2,2'-dioxo-1*H*,1'*H*-3,3' -bipyrrole-5,5'-dicarboxylate; *d,l*-**13** and *meso*-**13**, resp.) was synthesized according to a previously published procedure [31].

### 2.2. Isolation, Gram-Staining and DNA Genotypization of the Lactobacteria

Lactobacteria were isolated from the sediments of a fermentation batch, which was obtained from the vegetative parts of *M. perennis* and prepared according to a GHP protocol (German Homoeopathic Pharmacopoeia, procedure 34c) [32] after 3 months of incubation, by inoculation on MRS agar and incubation at 33 °C. Two different bacteria strains were segregated. Gram-staining of a smear of the bacteria strains was performed with an automated PREVI-Color Gram-staining system (bioMérieux Deutschland GmbH, Nürtingen, Germany) according to the manufacturer's specifications. For DNA genotyping, individual colonies of the pure strains were transferred into reaction tubes (Eppendorf, Germany) and stored at −80 °C. The genomic DNA of both bacteria was isolated by bead beating, phenol-CHCl$_3$ extraction, and *iso*-PrOH precipitation as described previously [33]. 16S rRNA genes of 1499 bp long were amplified from genomic DNA using the universal primer GM3

(5′-AGAGTTTGATCMTGGC-3′) and GM4 (5′-TACCTTGTTACGACTT-3′) in a polymerase chain reaction (PCR). The PCR was performed in 25 μL batches composed as follows: 2.5 μL dNTP (each 2.5 mM), 0.25 μL MgCl$_2$ (100 mM), 1 μL BSA (30 mg/mL), 15.2 μL H$_2$O, 0.25 μL of each primer (20 pmol, 1U Go Taq$^®$ G2 DNA polymerase (Promega, USA) and 5 μL Green Taq$^®$ reaction buffer. The PCR program consisted of an initial denaturation step at 95 °C for 5 min, 31 cycles at 95 °C for 1 min, 42 °C for 2 min and 72 °C for 1 min, with a final extension at 72 °C for 10 min. The PCR products were purified with a Peqlab Gold purification kit (VWR, Germany) according to the manual, and an aliquot was Sanger sequenced by GATC-Biotech (Eurofins Genomics GmbH, Ebersberg, Germany). Afterwards, sequences were aligned with the BioEdit program (Copyright$^®$ 2005, *T. Hall*, Ibis Therapeutics, Carlsbad, CA, USA) and compared with similar sequences of reference organisms using the BLAST function of the National Center for Biotechnology Information (NCBI) [34]. Finally, the analyzed DNA sequences were deposited at the NCBI GenBank [34]. For accession numbers see Table 1.

**Table 1.** Identification of the LAB strains, used in the present study, by 16S rRNA gene sequencing.

| Isolate (Strain) | GenBank Accession Number [1] | Sequence Length (bp) | Closest Relative in NCBI [2] | Similarity |
|---|---|---|---|---|
| LP1 | MK841313.1 | 1083 | *Lactobacillus plantarum* strain 2.7.17, MK611349.1 | 100% |
| PP1 | MK841045.1 | 1441 | *Pediococcus pentosaceus* strain CMGB-L16, MF348228.1 | 100% |

[1] Type material, accessed on April 26, 2019. [2] NCBI, National Center for Biotechnology Information [34].

### 2.3. Plant Material, Extraction (General Procedure)

Root parts of *M. perennis* were collected in September and aerial parts in March 2016, 2017 and 2018 in the mountain forest 565 m above sea level, close to Bad Boll/Eckwaelden (Germany), cleaned by rinsing with tap water and stored at −80 °C until investigation. Voucher specimens were deposited at the herbarium of the Institute of Botany, Hohenheim University (Germany), and the plant material was identified by Dr. R. *Duque-Thüs* (voucher number: HOH-020290). For extraction, roots and aerial parts of *M. perennis* (65 g) were immersed in water (420 mL) and bubbled with nitrogen (15 min). Subsequently, the plant material was minced for 3 min using an Ultra-turrax$^®$ (15,000 rpm; IKA-Werke GmbH and Co. KG, Staufen, Germany), again treated with N$_2$ and the slurry was allowed to stand for 24 h at +4 °C. Then, the plant material (sediment) was removed by vacuum suction over Celite.

### 2.4. Fermentation of Aqueous M. perennis Extracts with Isolated Lactobacteria Strains

Fermentations were performed in 50 mL or 500 mL batches. For this purpose, the filtrate was sterilized by filtration of the aqueous *M. perennis* extract over a sterile 0.2 μm cellulose acetate membrane (VWR International, Bruchsal, Germany). Afterwards, the filtrates were inoculated with overnight cultures of PP1 and LP1 in MRS broth (1 mL each) with a bacteria concentration of $2.6 \times 10^8$ cells/mL and $5.1 \times 10^8$ cells/mL, respectively, and incubated at 33 °C. For semi-quantitative investigations (stage monitoring) fermentation was performed in triplicate ($n = 3$). Culture broths were slightly agitated once a day to shake up precipitates. At $t = 0, 1, 3, 6$ and 14 days, samples of the fermentation broth (20 mL each) were withdrawn for GC-MS analysis and extracted with AcOEt ($2 \times 20$ mL). After drying (Na$_2$SO$_4$), the AcOEt extracts were filtered over a folded filter and the solvent was removed by vacuum rotary evaporation. The obtained residues were re-dissolved in 0.2 mL AcOEt and used for GC-MS analyses. For semi-quantitative determinations, the AcOEt extract was spiked with 1 mL of an *n*-eicosane AcOEt solution (0.025 mg/mL) prior to rotary evaporation. The increase of peak areas (*Atx*) was analyzed by GC-MS and calculated according to Equation (1):

$$\left( \frac{\left( \frac{Atx}{AStx} - \frac{At0}{ASt0} \right)}{ASt0} \right) \times 100 = increase\ in\ peak\ area\ \% \tag{1}$$

where *Atx* is the area of the analyte at a certain time, *At*0 is the area of the analyte at t = 0 days, *AStx* is the area of the internal standard at a certain time and Ast0 is the area of the internal standard at t = 0 days.

## 2.5. GC-MS Analyses

GC-MS analyses were performed with a PerkinElmer Clarus 500 gas chromatograph with split injection (split ratio 30:1, injection volume 1.0 µL) in conjunction with a single quadrupole mass spectrometer. The column and temperature program were used in accordance with a previous publication [31].

## 2.6. NMR Spectroscopy

NMR spectra were recorded in $CDCl_3$ or $CD_3OD$ at 500 or 600 ($^1H$) and 125 or 150 MHz ($^{13}C$) using a 500 MHz Varian Inova and a 600 MHz Bruker Avance III HD NMR spectrometer mounted with a BBO Prodigy cryo-probe. Chemical shifts are reported in δ [ppm] and refer to residual (non-deuterated) solvent signals of $CDCl_3$ ($^1H$: 7.27; $^{13}C$: 77.00 ppm) and $CD_3OD$ ($^1H$: 3.31; $^{13}C$: 49.20 ppm). For NMR spectra evaluation the program SpinWorks 3.1.7. (Copyright® 2010, *K. Marat*, University of Manitoba, USA) was applied.

## 2.7. Enrichment of Isochrysohermidinic Acid (**14**) from an Aqueous M. perennis Extract

Frozen roots of *M. perennis* (65.5 g) were extracted with water (420 mL) under nitrogen ($N_2$) atmosphere for 12 h as described above. After filtration, the obtained aqueous extract was kept for another 12 h in a refrigerator. For the isolation of isochrysohermidinic acid (**14**) the extract (385 mL) was applied onto an Amberlite® IRA 402 anion exchanger column (l = 25 cm, d = 3 cm), which was conditioned with 0.1 N HCl (1 L) and water (4 L). After extract application, the column was washed with water (2 L, fraction discarded) and the target material eluted with 0.1 N HCl (600 mL). Finally, the aqueous HCl was distilled off using a vacuum rotary evaporator to yield a brown hemi-crystalline syrup (0.61 g), which was kept at −30 °C until investigation.

## 2.8. RP-HPLC-(DAD)/ESI-MS^n Analyses

Liquid chromatographic / mass spectrometric analyses (LC-MS) were performed using an Agilent 1200 HPLC system (Agilent Technologies Inc., Palo Alto, USA) coupled to an HCTultra ion trap (Bruker Daltonik GmbH, Bremen, Germany) and an ESI source, controlled by Agilent Chemstation for LC 3D systems (Rev. B01.03SR1 (204)) and EsquireControl software (V7.1) [31]. A Kinetex® $C_{18}$ reversed-phase column (2.6 µm particle size, 150 × 2.1 mm i.d., Phenomenex Ltd., Aschaffenburg, Germany) was used for chromatographic separation at 25 °C and a flow rate of 0.21 mL/min. The mobile phase consisted of $HCOOH/H_2O$ 0.1:99.9 (*v/v*; eluent A) and MeCN (mobile phase B). The injection volume of each sample was 10 µL, and the gradient used was as follows: 0–10 min, 0%–10% B; 10–25 min, 10% B; 25–60 min, 10%–23 % B; 60–62 min, 23% B; 62–65 min, 23%–100% B; 65–70 min, 100 % B; 70–75 min, 100%–0% B; 75–85 min, 0% B. The MS system was operated in the positive ionization mode applying the following parameters: capillary voltage: −4000 V, dry gas ($N_2$) flow: 9.00 L/min with a capillary temperature of 365 °C; nebulizer pressure: 35 psi. Full scan mass spectra (*m/z* 50–1000) of HPLC eluates were recorded in the auto MS/MS mode during chromatographic separation. To obtain further structural information, collision induced dissociation (CID) experiments were performed by MS/MS fragmentation of $[M+H]^+$ ions.

## 2.9. Isolation of Plicatanin B (**8**) and N-Methylmaleimide (**5**) from a Fermentation Broth of M. perennis

A fermentation broth of *M. perennis* (1.37 L), prepared from aerial parts according to a GHP procedure [32] was extracted with AcOEt (4 × 300 mL), the AcOEt extract was dried ($Na_2SO_4$), filtered by vacuum suction and the solvent was removed by vacuum rotary evaporation to yield an olive

coloured syrup (1.25 g). The latter was applied to vacuum liquid chromatography (VLC), using a silica column (100 g SiO$_2$) and a CH$_2$Cl$_2$/MeOH gradient (from 100:0 to 95:5, *v/v*). Two fractions containing the target compounds were separated (identified by TLC), unified and dried by rotary evaporation yielding a residue of 0.17 g. Compounds **5** and **8** were isolated from the latter by centrifugally accelerated thin layer chromatography (CTLC, Chromatotron®, T-Squared Technology, San Bruno, CA, USA) [35] with a 2 mm layer (SiO$_2$/gypsum/fluorescence indicator 254 nm, 45:18:1.2 (*w/w/w*)), applying an *n*-hexane/AcOEt gradient from 100:0 to 80:20 (*v/v*). The pure compounds **5** (31.3 mg) and **8** (7.5 mg) were obtained from the corresponding fractions after solvent removal.

*2.10. Synthesis of Reference Compounds*

2.10.1. Synthesis of Plicatanin B (bis-(3-methoxy-1-*N*-methyl-maleimide); **8**)

A solution of *rac*-isochrysohermidin (**13**, 0.177 g, 0.444 mmol) in THF/MeOH (1:1, *v/v*, 40 mL) was treated under N$_2$ atmosphere with a 0.25 M aqueous NaOH solution (8 mL). After stirring at room temperature (22 h) the solvent was removed by vacuum rotoevaporation (T = 38 °C, *p* < 22 mbar). The remaining residue was dissolved in water (120 mL) and the pH adjusted to 7.0 by addition of AcOH. Subsequently, 20 mL of a freshly prepared aqueous 0.62 % bromine solution (1 mL, 3.12 g Br$_2$ dissolved by sonication in 0.5 L water) was added and the reaction mixture stirred for 7 h at room temperature, while the color turned from orange into yellow-green. Subsequently, the aqueous solution was saturated with NaCl and extracted with AcOEt (3 × 100 mL). The combined extracts were dried (Na$_2$SO$_4$) and the solvent distilled off to yield the crude product (0.085 g). The latter was purified by CTLC (Chromatotron®, conditions see above). The elution of the target product was performed with a hexane/AcOEt gradient from 100:0 to 60:40 (*v/v*). Yield: 0.0508 g (40.8 % of the theory). R$_f$ (SiO$_2$; CH$_2$Cl$_2$/MeOH 19.5:0.5 (*v/v*)) 0.51. GC/MS purity: 98.5% (at retention time 36.1 min), GC/MS data see Table 2. UV/VIS (MeCN): 235 (4.36), 350 (3.18). $^1$H-NMR (CDCl$_3$, 600 MHz): 4.14 (s, H-(C7, 7')), 3.06 (s, H-(C6, 6')). $^{13}$C-NMR (CDCl$_3$, 150 MHz): 169.57 (C(2, 2')), 164.93 (C(5, 5')), 156.70 (C(3, 3')), 99.69 (C(4, 4')), 59.71 (C(7, 7')), 24.10 (C(6, 6')). The NMR shifts (see also Supplementary Material, Figures S5 and S6) are in agreement with literature data [36].

2.10.2. Synthesis of 3-Bromo-1*N*-methylmaleimide

The compound was synthesized according to a modified literature procedure [37]. In brief, a solution of *N*-methylmaleimide (2.50 g, 22.50 mmol) in MeOH (110 mL) was treated dropwise with bromine (Br$_2$, 2.6 mL, 8.11 g, 50.75 mmol) and stirred for 24 h at room temperature Then, the solvent was removed by vacuum rotoevaporation. To remove Br$_2$, MeOH (2 × 50 mL) was added and distilled off in vacuo. The obtained orange residue was dissolved in THF (100 mL), Et$_3$N (5.4 mL, 3.94 g, 38.94 mmol) was added dropwise under stirring and the reaction mixture was stirred for further 10 h at room temperature Afterwards, the precipitated ammonium salts were filtered off by vacuum suction, washed with THF (2 × 50 mL), and the solvent was removed from the filtrate by rotoevaporation. The crude product (5.1 g) was purified by vacuum liquid chromatography (VLC) on silica using an *n*-hexane/AcOEt gradient (from 100:0 to 70:30, *v/v*). The corresponding fractions containing the target product (identified by TLC) were combined, the solvent was distilled off and the obtained yellow solid residue was washed with *n*-hexane (2 × 50 mL). The pure compound was obtained after recrystallization from *n*-hexane. Yield: 2.90 g (67.8 % of the theory). Mp. 90.5 °C, Mp.$_{Lit.}$ 77–79 °C [37]. R$_f$ (SiO$_2$; CH$_2$Cl$_2$/MeOH 19.5:0.5 (*v/v*)) 0.68. UV/VIS (MeCN): 238 (4.17). GC/MS purity (70 eV): >99.9 % (retention time 9.9 min): *m/z* 191, 189 (59/61) [M$^+$($^{81}$Br/$^{79}$Br)], 163, 162 (3/4), 134, 132 (42/44), 110 (8), 106, 104 (35/36), 82 (18), 66 (8), 56 (14), 53 (100).

**Table 2.** Gas chromatography-mass spectrometry (GC-MS) data of compounds identified in unfermented and fermented root extracts of *M. perennis.*

| Compound | Retention Time (min) | Characteristic Mass Signals, *m/z* (% BPI) |
|---|---|---|
| mequinol (4-methoxyphenol; **9**) | 11.1 | 124 (M⁺, 100), 109 (98), 95 (2), 81 (46), 65 (5), 53 (15) [1] |
| 3-methoxy-1*N*-methyl-maleimide (**5**) | 11.9 | 141 (M⁺, 23), 123 (8), 113 (12), 112 (25), 84 (23), 69 (100), 56 (9), 53 (7) |
| (−)-*cis*-myrtanol (**10**) | 12.8 | 136 ([M-H₂O]⁺, 11), 123 (68), 121 (42), 107 (15), 95 (28), 93 (100), 81 (65), 79 (38), 69 (73), 67 (80), 55 (45) [1,2] |
| 3-methylhermidin (**6**) | 16.6 | 185 (M⁺, 100), 170 (65), 156 (68), 142 (30), 128 (20), 126 (15), 114 (13), 99 (17), 83 (40), 68 (74), 58 (64) |
| 3,4-dimethoxyphenol (**11**) | 17.0 | 154 (M⁺, 100), 139 (96), 125 (4), 111 (48), 96 (11), 93 (27), 81 (9), 69 (16), 65 (18), 55 (18) [1] |
| 3-methylhermidin quinone (**7**) | 18.7 | 183 (M⁺, 52), 154 (1), 137 (2), 126 (6), 112 (3), 98 (23), 83 (100), 70 (8), 55 (8) |
| hermidin quinone (**3**) | 20.9 | 169 (M⁺, 35), 142 (0.4), 112 (35), 84 (21), 69 (100), 56 (7), 53 (9) [1] |
| plicatanin B (bis-(3-methoxy-1*N*-methyl-maleimide); **8**) | 35.8 | 280 (M⁺, 100), 265 (20), 251 (24), 237 (11), 208 (17), 195 (6), 180 (35), 178 (22), 166 (21), 152 (20), 123 (45), 109 (5), 95 (72), 80 (37), 72 (16), 67 (8), 53 (6) |
| 3-(3-methoxy-1*N*-methyl-malimido)-hermidin quinone (**12**) | 40.0 | 308 (M⁺, 67), 280 (54), 265 (14), 251 (27), 237 (13), 223 (11), 208 (72), 195 (34), 180 (76), 166 (13), 152 (30), 123 (62), 109 (8), 95 (100), 80 (57), 69 (10), 53 (11) |
| chrysohermidin (bis-(hermidin quinone); **4**) | 43.9 | 336 (M⁺, 9), 321 (74), 308 (12), 280 (53), 265 (23), 251 (23), 236 (76), 208 (62), 195 (25), 180 (100), 167 (11), 152 (37), 123 (52), 111 (13), 95 (94), 80 (58), 68 (13), 53 (11) [1] |

[1] MS data in agreement with [26]. [2] M⁺ not observed.

### 2.10.3. Synthesis of 3-Methoxy-1*N*-methylmaleimide (**5**)

3-Bromo-1*N*-methylmaleimide (1.00 g, 5.263 mmol) and dibenzo-18-crown-6 (0.197 g, 0.547 mmol) were dissolved in MeCN (30 mL), stirred for 10 min and KOMe (0.40 g, 5.704 mmol) was added. After stirring in an oil bath (40 °C, 3 h), the reaction was quenched by addition of saturated aqueous $NH_4Cl$ solution (prepared from 30 g $NH_4Cl$ and 90 mL water). Thereafter, the mixture was extracted with $CHCl_3$ (4 × 50 mL), the combined $CHCl_3$ extracts were dried ($Na_2SO_4$) and the solvent removed by vacuum rotoevaporation to yield a crude residue (1.15 g). The pure compound **5** was obtained from the latter by repeated VLC on silica 60 G (2 × 60 g $SiO_2$), utilizing an *n*-hexane/AcOEt gradient from 100:0 to 60:40 (*v/v*). The fractions containing **5** (detected by TLC) were unified, the solvent was removed by rotoevaporation, and the product was dried in vacuo. Yield: 0.128 g (17.2 % of the theory), white solid. $R_f$ ($SiO_2$; $CH_2Cl_2$/MeOH 19.5:0.5 (*v/v*)) 0.57. GC/MS purity (70 eV): 91 % (retention time 11.9 min): For MS data see Table 2. $^1$H-NMR ($CDCl_3$, 500 MHz): 5.41 (s, H-(C4)), 3.93 (s, H-(C7)), 2.99 (s, H-(C6)).$^{13}$C-NMR ($CDCl_3$, 125 MHz): 170.19 (C(2)), 165.65 (C(5)), 161.07 (C(3)), 96.19 (C(4)), 58.85 (C(7)), 23.44 (C(6)). The NMR shifts (see also Supplementary Material, Figures S3 and S4) are in agreement with literature data [38].

### 2.10.4. Synthesis of Methylarbutin (4-Methoxyphenol-*β*-D-glucopyranoside; **15**)

Arbutin (4-hydroxyphenol-*β*-D-glucoside, 16.74 g, 61.50 mmol) was dissolved in aqueous 1% KOH solution (*w/v*, 430 mL, 76.64 mmol) under $N_2$ atmosphere. Then, dimethylsulfate (5.85 mL, 7.78 g, 61.68 mmol) was added dropwise and the mixture was stirred 1 h at room temperature and further 1 h at 110 °C in an oil bath. After cooling to room temperature the reaction mixture was extracted with AcOEt (3 × 430 mL) and the combined AcOEt extracts were dried ($Na_2SO_4$). After filtration, the solvent was removed in vacuo to yield the crude product (2.42 g, 13.7 % of the theory). The latter was recrystallized from AcOEt. White solid. M.p. 172 °C (M.p.$_{Lit}$ = 170 °C [39]). $R_f$ ($SiO_2$; MeOH/AcOEt/$H_2O$ 3:15:2 (*v/v/v*)) 0.47. UV/VIS (MeCN): 225 (4.07), 285 (3.42). GC-MS purity (as TMS derivative, 70 eV): >90% (retention time 43.5 min), impurity: arbutin; *m/z* 469 (1), 450 (7), 361 (43), 331 (4), 319 (5), 271 (13), 243 (12), 217 (33), 196 (92), 191 (8), 169 (18), 147 (34), 129 (19), 117 (7), 103 (16), 73 (100). $^1$H-NMR ($CD_3OD$, 600 MHz): 7.05 (*d*, $J_{H2,6/H3,5}$ 9.1, H-(C2, 6)), 6.83 (*d*, $J_{H3,5/H2,6}$ 9.1 , H-(C3, 5)), 4.77 (*d*, $J_{H3,5/H2,6}$ 7.5, H-(C1′)), 3.88 (*dd*, $J_{Ha-6'/Hb-6'}$ 12.0, $J_{Ha-6'/H-5'}$ 1.8, Ha-(C6′)), 3.75 (*s*, H-(C7), $OCH_3$), 3.70 (*dd*, $J_{Hb-6'/Ha-6'}$ 12.0, $J_{Hb-6'/H-5'}$ 5.3, Hb-(C6′)), 3.45–3.36 (*m*, H-(C2′, 3′, 4′, 5′)). $^{13}$C-NMR ($CD_3OD$, 150 MHz): 156.84 (C(4)), 153.43 (C(1)), 119.41 (C(2, 6)), 115.64 (C(3, 5)), 103,67 (C(1′)), 78.29 (C(3′)), 78.21 (C(5′)), 75.18 (C(2′)), 71,63 (C(4′)), 62.76 (C(6′)), 56.24 (C(7)). The NMR data (see also Supplementary Materials, Figures S7 and S8) are in agreement with literature data [39,40].

## 3. Results and Discussion

### 3.1. Identification of the Lactobacteria in Fermented Aqueous *M. perennis* Extracts

Since the LAB microflora of fermented *M. perennis* extracts has not been studied so far, we isolated two bacterial strains as pure cultures from sediments of a fermentation broth after 3 months of incubation. By microscopic examination, it was demonstrated that both bacterial strains were Gram-(+), while one strain consisted of cocci and the other of rod-shaped bacteria (bacilli).

The isolated strains LP1 (MK841313) and PP1 (MK841045) were identified by 16S rRNA gene sequence analysis (Table 1) as members of the genus *Lactobacillus* and *Pediococcus*, respectively [41,42]. The closest relatives show a 100% similarity with *Lactobacillus plantarum* and *Pediococcus pentosaceus* (Table 1). Both bacteria are typical homo-fermentative representatives of the LAB flora [1,43] and often occur in fermented plant materials [41,44].

### 3.2. Fermentation of M. perennis with the Lactobacteria Isolates and GC-MS Analysis of the Metabolites

To study the bioconversion of plant metabolites in a fermentation model, roots of *M. perennis* were used in a first step because of high hermidin (**1**) content compared to herbal parts containing lower

amounts of this alkaloid [26]. Hermidin (**1**) is an oxygen-sensitive compound which is only stable under reducing physiological conditions. In the presence of oxygen, **1** is easily oxidized forming the blue anionic radical cyanohermidin (**2**) (Figure 1). This occurs when plant cell integrity is damaged, e.g., after harvest or upon plant extraction [24,25]. Consequently, **2** further reacts forming the oxidation products hermidin quinone (**3**) and chrysohermidin (**4**), which display multifaceted downstream reactions in aqueous solvents [31]. In the present study *M. perennis* roots, and in a second step also herbal parts, were macerated with water under nitrogen (N$_2$) atmosphere to prevent oxidation reactions of the plant constituents.

**Figure 1.** Proposed metabolic pathway of hermidin (**1**). Conversion by oxidation (*a–c*) and catabolization (*e–d*) upon the action of LAB. (*a*) Deprotonation and oxidation (-2H$^+$; –e$^-$). (*b*) Oxidation (–e$^-$). (*c*) Dimerization and protonation (+2H$^+$). (*d*) Benzilic acid rearrangement and oxidative decarboxylation (–CO$_2$, –2H$^+$). (*e*) Partial six-ring degradation.

After sterile filtration, the obtained aqueous extracts (pH 7.2) were incubated either with the LAB strain PP1 or LP1. The pH of the fermentation broths readily declined to 3.8–3.5 after 1 to 2 days of incubation, indicating the formation of LA, while from this point on the pH remained constant over the entire fermentation period. After time intervals of *t* = 0, 1, 3, 6 and 14 days samples were withdrawn, extracted with AcOEt and investigated by GC-MS (EI mode). In the AcOEt extracts of an unfermented aqueous sample (*t* = 0 days, Figure 2A) several known compounds, like mequinol (4-methoxyphenol; **9**, *m/z* 124), (−)-*cis*-myrtanol (**10**, *m/z* 154), 3,4-dimethoxyphenol (**11**, *m/z* 154), hermidin quinone (**3**, *m/z* 169) and chrysohermidin (**4**, *m/z* 336), were detected by GC-MS at retention time 11.1, 12.8, 17.0, 20.9, and 43.9 min, respectively, and assigned based on their mass spectra and comparison with previously published data (Figure 1, Figure 2A and Table 2) [26]. Already after 1 day of fermentation the monomeric hermidin quinone (**3**) and the dimeric chrysohermidin (**4**) were almost completely degraded, while two novel metabolites at retention time 16.6 and 18.7 min were analyzed by GC-MS (Figure 2B, exemplarily shown for a PP1 culture broth, while inoculation with LP1 revealed a similar pattern).

The odd-numbered molecular ion peaks at *m/z* 185 and 183 and distinct fragmentation patterns [M–MeN]$^+$ and [M–MeN–CO]$^+$ (Table 2) indicate these compounds to be methoxy-*N*-methylpiperidione derivatives, amplified by one C1 unit. Based on mass spectral similarities to literature data of ethylhermidin and ethylhermidin quinone (Mw 199 and 197 Da, respectively), formerly detected in *M. perennis* extracts fermented with a mixed LAB flora [30], we conclude that the novel constituents are methylhermidin (**6**) and methylhermidin quinone (**7**) (Figure 1). Starting from hermidin (**1**), the spontaneously formed oxidation products **3** and **4** (Figure 1) are supposedly converted by enzymes like decarboxylases. One pathway leads to the degradation of one six-membered ring of chrysohermidin

(**4**) to form the methylhermidin derivatives **6** and **7**. The formation of these latter compounds **6** and **7** by isolated LAB strains, as observed in this study, is different from previously published results, where *M. perennis* extracts were fermented by a spontaneously developed LAB flora. In contrast, spontaneous fermentation due to the wild microbial flora converts **4** into ethylhermidins [30]. One reason might be the different activities of the enzymes of a mixed LAB microflora compared to isolated strains. Interestingly, the previously reported metabolites ethylhermidin and ethylhermidin quinone were found in the current study only in trace amounts in fermentation batches with PP1 and LP1.

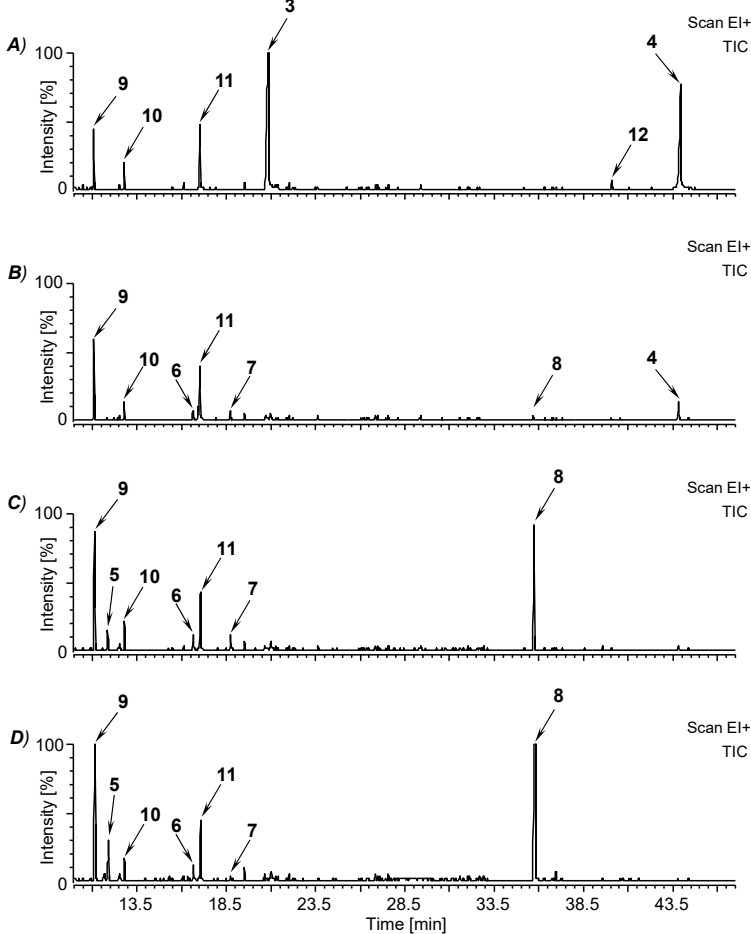

**Figure 2.** Sections of GC-MS profiles (EI) showing metabolites of *M. perennis* root extracts. (**A**) Before (*t* = 0 days); (**B**) after 1 day (**C**); after 6 days and (**D**) after 14 days of fermentation with *Pediococcus* sp. (PP1). For compound assignment, see Table 2.

In the course of fermentation, after 6 days, a prominent peak at retention time 35.8 min appeared (Figure 2C). This compound exhibited a molecular ion $[M]^+$ at *m/z* 280 (Table 2) in the EI mass spectrum but could not be readily assigned.

The fragmentation pattern of this molecule was similar to that of chrysohermidin **4** (Table 2), and the even numbered $[M]^+$ also points to a dimeric compound, containing two nitrogen atoms. From a strong yellow-green fluorescent spot, which was observed in the TLC of the AcOEt extract (excitation wavelength: 366 nm, see Supplementary Materials, Figures S1 and S2) and which correlated with the compound, this constituent is concluded to carry a UV fluorescent chromophore, probably a maleimide [45]. For unequivocal structure assignment, an attempt was made to isolate this particular compound. By EtOAc extraction of a fermentation broth, obtained by fermentation of herbal parts from *M. perennis* in accordance with a GHP (German Homoeopathic Pharmacopoeia) literature procedure [32], a fraction was yielded containing the target analyte. The pure compound was recovered

(yield: 5.4 mg/L fermentation broth) by subsequent purification via vacuum liquid chromatography (VLC) and centrifugally accelerated thin layer chromatography (CTLC) [35] on silica. The $^1$H-NMR spectrum of the pure isolated compound revealed only two different spin systems: a MeN group ($\delta$ (H) 4.14 (*s*)) and a MeO group ($\delta$ (H) 3.05 (*s*)). Based on mass spectral, $^1$H and $^{13}$C NMR data (see Supplementary Materials, Figures S5 and S6) and comparison with the literature, the compound was identified as plicatanin B (bis-(3-methoxy-1*N*-methylmaleimide); **8**), which had previously been isolated from *Chrozophora plicata* (Euphorbiaceae family) [36]. An $\alpha$-glucosidase inhibitory activity was detected in vitro for compound **8** by the same authors [36].

It is particularly noteworthy that in the course of our investigation also the monomer **5** (3-methoxy-1*N*-maleimide, Mw 141 Da) was obtained from the same fermentation broth. It was detected after TLC separation as a light-blue fluorescent spot (Supplementary Materials, Figures S1 and S2) and in GC-MS analyses as a peak at retention time 11.9 min. Further, **5** was identified by comparing mass spectral data with those of a synthesized reference compound. However, it was found in lower amounts in *M. perennis* root extracts (Figure 2C,D), fermented with pure cultures of PP1 or LP1, as compared to the dimer plicatanin B (**8**). When comparing GC-MS chromatograms of root extracts fermented either with PP1 or LP1, both were not significantly different regarding the compound pattern (Supplementary Materials, Figure S9), thus revealing similar enzymatic activities of both LABs. However, differences were observed in fermented aqueous extracts obtained from root or herbal parts. The latter tended to show higher concentrations of **8** as deduced from GC-MS analyses (Supplementary Materials, Figure S9).

Moreover, for corroborating structure assignment and to gain further spectroscopic information **8** was synthesized starting from isochrysohermidin (**13**, Figure 3). For this purpose, **13** [31] was treated in a THF/MeOH solution with 0.25 N NaOH. After pH adjustment to 7.0 with AcOH, the intermediate isochrysohermidinic acid (**14**) thus formed, was decarboxylated upon addition of aqueous bromine (Br$_2$) solution according to a method of Pink and Stewart [46].

**Figure 3.** Synthesis of plicatanin B (**8**) by conversion of isochrysohermidin (**13**).

After AcOEt extraction and chromatographic purification, plicatanin B (**8**) was obtained as a pure compound with a 40.8% yield. The chromatographic and spectroscopic features (GC-MS, LC-MS$^n$, NMR) of synthesized **8** were identical to those of the natural constituent obtained upon fermentation. To the best of our knowledge, a total synthesis of **8** was hereby accomplished for the first time. In addition to compound **8**, GC-MS chromatograms (Figure 2A–D) revealed the formation of mequinol (4-methoxyphenol; **9**, retention time 11.1 min) upon fermentation. The increase of both compounds was semi-quantitated by evaluation of peak areas of the two analytes **8** and **9** at $t = 0, 1, 3, 6$ days (Figure 4). Even though a comparatively short time period of 6 days was monitored, it was found that the formation of **9** was slightly delayed compared to that of **8** (Figure 4). Comparable findings were also deduced from TLC experiments performed upon fermentation (Supplementary Materials, Figure S2).

Since a precursor molecule of **9** could not be captured by GC-MS, its formation by hydrolytic cleavage of a glycosidic derivative was assumed. To prove this assumption, an AcOEt extract of an unfermented aqueous root extract was silylated together with a methylarbutin (4-methoxyphenol-$\beta$-D-glucoside; **15**) reference compound. The latter was synthesized from arbutin (4-hydroxyphenol-$\beta$-D-glucoside) by methylation with dimethylsulfate and investigated via GC-MS (Figure 5). The retention time and EI mass spectrum of the silylated reference compound (methylarbutin-TMS; **15**-TMS) could be aligned with the natural constituent (Figure 5). Even though methylarbutin (**15**)

was known before as a plant constituent [47–49], its occurrence in the roots of *M. perennis* has been proven here for the first time. Thus, the cleavage of **15** by LAB *β*-glucosidases (Figure 6) is a conclusive explanation for the increase in the concentration of mequinol (**9**) upon fermentation. Similar de-glycosylation reactions of plant constituents through the action of microbial enzymes are known from numerous investigations reported in the literature [11,12,50].

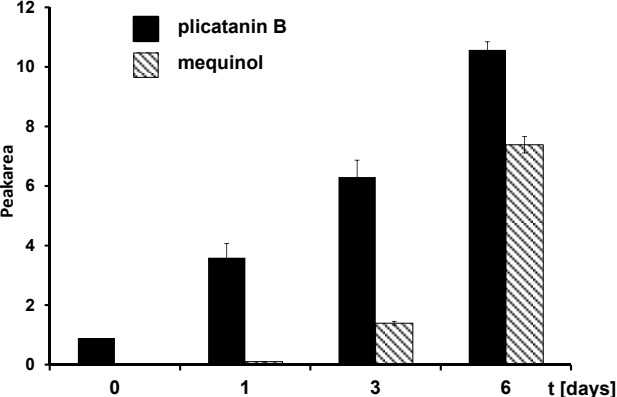

**Figure 4.** Increase of the peak areas of plicatanin B (**8**) and mequinol (**9**) upon fermentation of *M. perennis* aqueous root extracts with *Pediococcus* sp. (PP1). Peak areas were measured by GC-MS analysis of AcOEt extracts and calculated based on the internal standard eicosane (*n*-C$_{20}$); STD (*n* = 3).

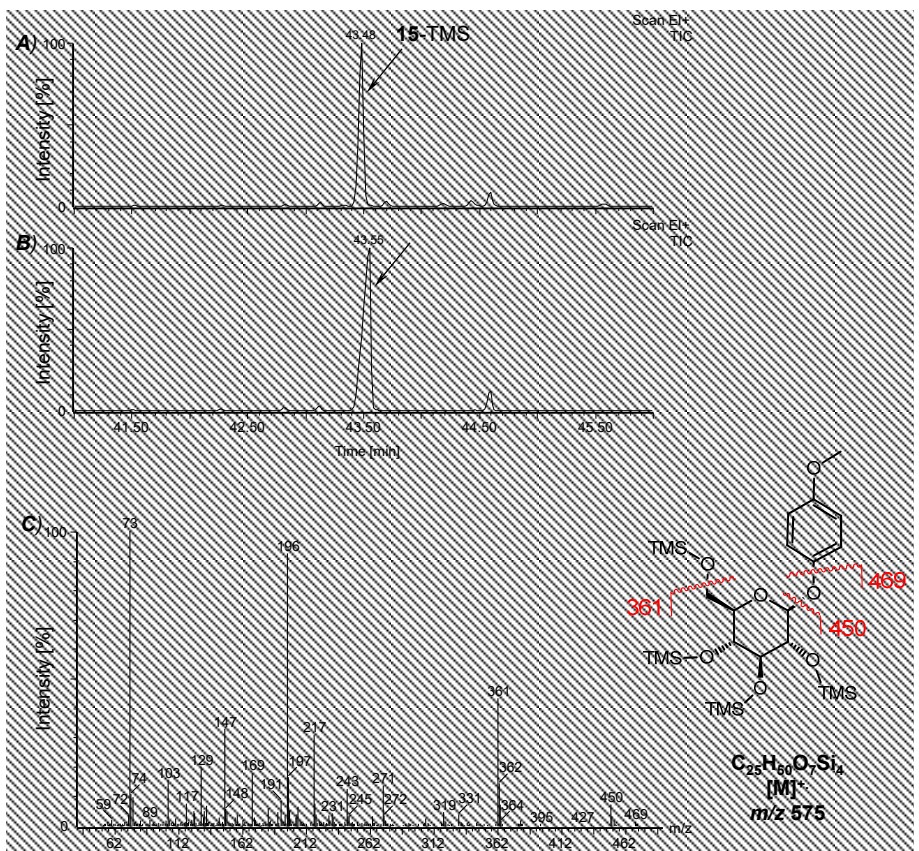

**Figure 5.** Detection of methylarbutin (**15**) in an unfermented aqueous extract of *M. perennis* roots. (**A**) Section of a GC-MS profile (EI) of an AcOEt extract after silylation. (**B**) GC-MS of a silylated reference standard. (**C**) EI mass spectrum of **15**-TMS. The red wavy lines illustrate MS fragmentation positions.

**Figure 6.** Enzymatic release of mequinol (**9**) from methylarbutin (**15**) by *Lactobacteria.*

### 3.3. Approach to the Reaction Mechanism Converting Chrysohermidin (*4*) into Plicatanin B (*8*)

The bioformation of plicatanin B (**8**) upon LAB fermentation is still not understood from a mechanistic point of view. We could recently demonstrate that chrysohermidin (**4**), as a dimeric piperidine-2,6-dione, spontaneously reacts in the presence of nucleophiles like methanol via benzilic acid rearrangement, under contraction of both six-membered rings into five-membered rings and esterification to yield isochrysohermidin (**13**, for structure, see Figure 3) [31]. Hence, a similar reaction of **4** in the presence of water was supposed, which should lead to isochrysohermidinic acid (**14**) as a further intermediate (Figure 7). The acid **14** might henceforth react to **8** by step-wise decarboxylation under the influence of microbial enzymes (Figure 7).

**Figure 7.** Proposed metabolic pathway of a step-wise conversion of chrysohermidin (**4**) into plicatanin B (**8**). (*a*) Benzilic acid type rearrangement and (*b*) decarboxylation. The intermediates **14** and **16** were detected by LC-MS, while **12** was analyzed by GC-MS (see Figure 2A).

Since **14** has not been described in the literature so far, its enrichment from an aqueous extract of *M. perennis* roots by binding onto an anion exchanger (Cl⁻ form) was performed as a first step. Subsequently, sugars and salts were removed by rinsing the exchanger resin with water. Compound elution and enrichment were achieved with 0.1 N HCl and subsequent removal of the aqueous HCl in vacuo. The obtained fraction was analyzed by $C_{18}$-RP-HPLC-MS$^n$ (Figure 8), which shows that the target molecule **14** was enriched by applying the aforementioned steps. The base peak chromatogram revealed a broad peak at low retention time (2.5–2.8 min) with a supposed molecular ion [M+H]$^+$ at *m/z* 373 (compound **14, Figure 8**A,B). This compound did not exhibit a characteristic UV adsorption (data not shown). Thus, structure assignment was based on its mass spectral behavior. Collision-induced dissociation (CID) showed the twofold neutral loss of $CO_2$ and $CH_3OH$ units from the molecular ion [M+H]$^+$, yielding fragments at *m/z* 297 and 221 (Figures 8C and 9). Subsequent fragmentation revealed the loss of a water molecule from the daughter ion at *m/z* 297 yielding a fragment at *m/z* 279, followed

by a twofold release of methylisocyanate [Me-N=C=O] from the fragment at *m/z* 221, forming ions at *m/z* 164 and 79, respectively (Figure 9).

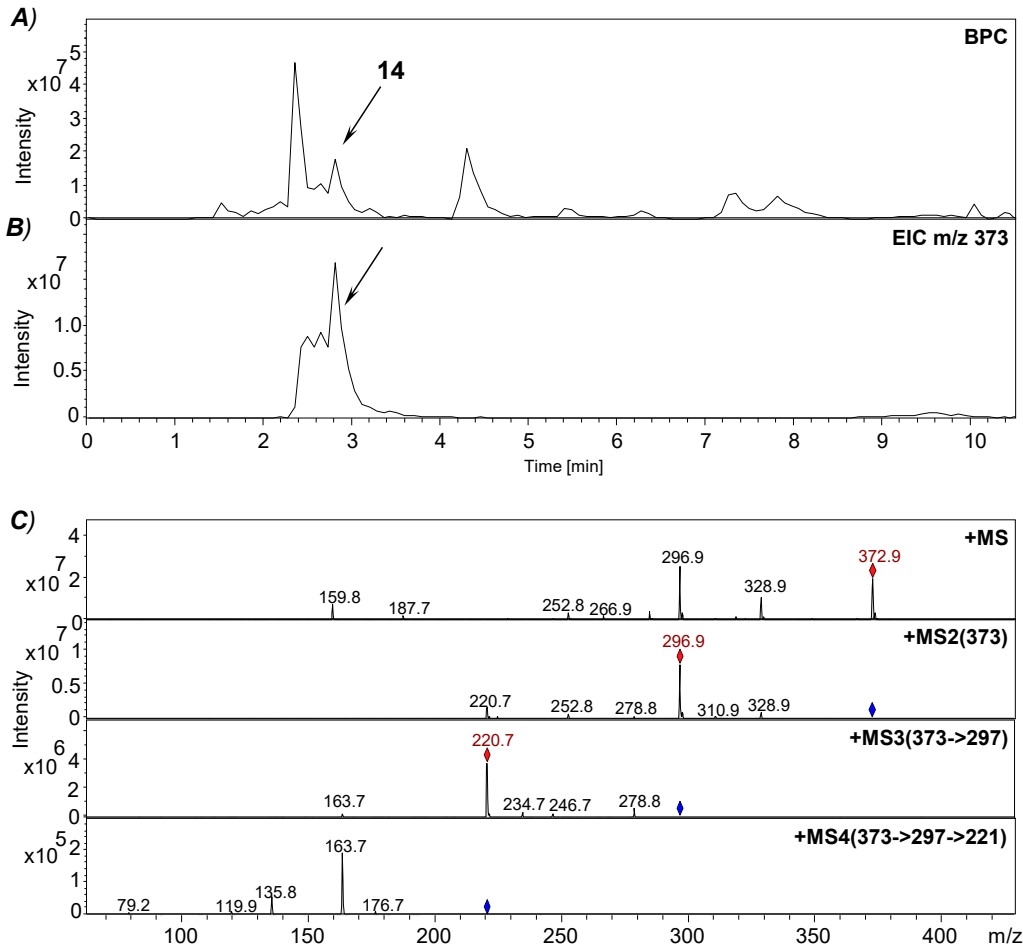

**Figure 8.** Detection of isochrysohermidinic acid (**14**) after enrichment from an unfermented aqueous *M. perennis* root extract by anion exchange chromatography. (**A**) Section of an LC-MS profile (BPC, positive ionization mode). (**B**) Extracted ion chromatogram (EIC) for [M+H]$^+$ at *m/z* 373. (**C**) Fragmentation of **14** ($C_{14}H_{16}N_2O_{10}$; Mw 372.28 Da).

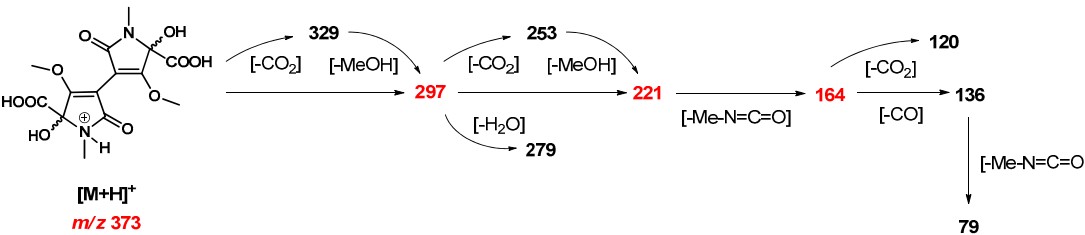

**Figure 9.** Proposed fragmentation of **14** upon CID (according to Figure 8C). Red numbers indicate the most intense fragments.

Even though attempts to isolate **14** from an aqueous extract of *M. perennis* failed and its total synthesis was not successful, the LC-MS data strongly support the aforementioned structure assignment.

Furthermore, plicatanin B (**8**) and the proposed intermediates **14** and **16** (Figure 7) were analyzed in the unfermented and fermented aqueous root extract of *M. perennis* by LC-MS, applying single ion extraction of the respective molecular ions [M+H]$^+$ (Figure 10A,B, retention time 2.3 and 13.5 min). In addition, **8** (retention time 40.6 min) was assigned by a reference standard (data not shown). Moreover, the proposed intermediate **12** (Figure 7) could be detected as a small peak at retention time

40.0 min in the GC-MS chromatogram of an AcOEt extract obtained from the unfermented aqueous root extract (Figure 2A, Table 2).

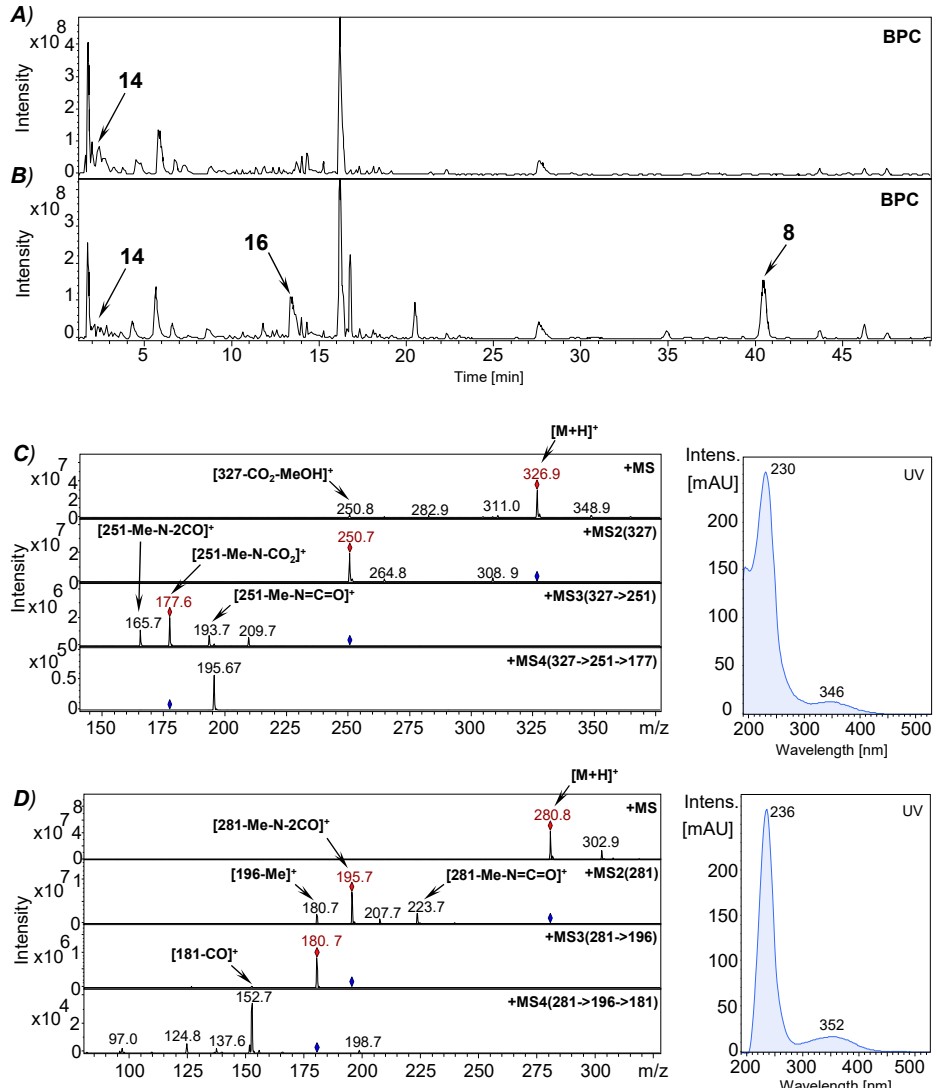

**Figure 10.** Detection of isochrysohermidinic acid (**14**), intermediate **16** and plicatanin B (**8**) in an unfermented and *Pediococcus* sp. (PP1) fermented aqueous extract from *M. perennis* roots. (**A**) Section of an LC-MS profile (BPC, positive ionization mode) of the unfermented extract. (**B**) LC-MS profile of the fermented extract (14 d). (**C**) MS$^n$ data of **16** ($C_{13}H_{14}N_2O_8$; Mw 326.26 Da). (**D**) MS$^n$ data of **8** ($C_{12}H_{12}N_2O_6$; Mw 280.23 Da). UV spectra of the respective compounds are illustrated on the right.

The mass spectrometric behavior of the proposed intermediate **16** (Figure 10C) was similar to that of **14**, i.e., it showed the release of a [$CO_2$-MeOH] unit from the molecular ion [M+H]$^+$ at *m/z* 327, yielding a fragment at *m/z* 251, being characteristic of a pyrrolidone-1-hydroxycarboxylic acid unit (Figure 10C). The latter showed the subsequent release of [Me-N=C=O], [Me-N-$CO_2$] and [Me-N-2CO] units from the *N*-methyl-maleimide core structure yielding fragments at *m/z* 194, 178 and 166, respectively. The latter fragmentation was also observed for **8** (Figure 10D). Further, the UV spectra of **8** and **16** (Figure 10) are characteristic of maleimides [45], thus corroborating structure assignment. The detected intermediates **12**, **14** and **16** support the proposed reaction mechanism (Figure 7). Starting from chrysohermidin (**4**), a concerted reaction path consisting of benzilic acid rearrangements, which may spontaneously proceed in the presence of water [31], and subsequent conversion of the pyrrolidone-1-hydroxycarboxylic acid units in the presence of LAB decarboxylases,

finally yields plicatanin B (**8**). The high decarboxylase activity of LAB cultures is known from the literature [51,52]. As a result, further plant constituents such as coumaric or caffeic acid derivatives are converted by LAB phenolic acid decarboxylases to form vinylphenols or ethylphenols [53–55].

## 4. Conclusions

Fermentation of medicinal plant constituents is still poorly described in the literature. The fermentation model used in the present study allows detailed investigations of the conversion of individual plant metabolites due to the activity of isolated microorganisms. We could demonstrate that the incubation of aqueous extracts with homo-fermentative *Lactobacillus* sp. (LP1) and *Pediococcus* sp. (PP1) follows a complex conversion route of the genuine plant constituents, which is due to the versatile set of LAB enzymes. In comparison, spontaneous fermentation with a wild bacterial flora expectedly yields preparations with higher variance in their chemical profiles. Results from such a fermentation model lay a general basis for the production of tailored fermented extracts from *M. perennis* but also from other medicinal plants with isolated Lactobacteria strains. Therefore, the application of starter cultures in medicinal plant fermentation practice may help to speed up or improve the control of fermentation processes, aiming at the standardization of extracts containing pharmaceutically active components. Moreover, the use of Lactobacteria opens up the possibility of a "green chemistry" process leading to isolated active ingredients like plicatanin B (8), which are otherwise costly to obtain by chemical total synthesis. Future studies may show a correlation between the bioactivity of such fermented extracts and the applied fermentation process.

**Supplementary Materials:** The supplementary materials are available online at http://www.mdpi.com/2311-5637/5/2/42/s1.

**Author Contributions:** Design of the study: P.L., D.R.K., F.C.S.; preparation of extracts, fermentation experiments, synthesis of reference compounds: P.L., M.B.; data acquisition: P.L., M.B.; NMR data interpretation: P.L., J.C.; DNA genotypization: S.S.; evaluation of the data and preparation of the manuscript: P.L., M.B., S.S., D.R.K., and F.C.S.

**Acknowledgments:** The authors want to thank *Dorothee Hagemann* (WALA Heilmittel GmbH) for preparing fermented aqueous extracts according to GHP protocols. We are also grateful to *Regina Herrmann, Monika Pickl* and *Cornelia Rechert* (WALA Heilmittel GmbH) for assistance in the cultivation of the microorganisms. *Rhinaixa Duque-Thüs* (Institute of Botany, Hohenheim University) is acknowledged for the identification of the plant specimens. We thank *Mario Wolf* (Institute of Chemistry, Hohenheim University) for assistance in NMR measurements, and *Jennifer Baer-Engel* (WALA Heilmittel GmbH) is gratefully acknowledged for proofreading of the manuscript.

**Conflicts of Interest:** The authors declare no conflict of interest.

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
