# Peer review of "Conversion of Plant Secondary Metabolites upon Fermentation of Mercurialis perennis L. Extracts with two Lactobacteria Strains"

_fermentation, doi:10.3390/fermentation5020042_

Reviewer 1 Report

On the manuscript entitled "Conversion of Plant Secondary Metabolites upon Fermentation of Mercurialis perennis L. Extracts with two Lactobacteria Strains” for Fermentation

The subject is of a high interest and in the scope of Fermentation and could be considered for publication in this journal but the minor revision of the manuscript is required.

The section with Methods should be replaced after the Introduction section, and not at the end of the manuscript.

Author Response

The section with Methods should be replaced after the Introduction section, and not at the end of the manuscript.

Response: Done! The "Materials & Methods" section have been shifted.

Reviewer 2 Report

Dear Authors,

In my opinion your article is very interesting, includes many new scientific facts and can be helpfull for other scientists to understand what is happening during described process.

Anyway I have some suggestions/comments:

- In my opinion the Introduction section should be shortened and what is more important you should include POISONING aspects of Mercurialis perennis. It is know that this plant can couse hard poisoninf effects, including death. I think that it should be describe and strongly highlithed in Introduction

- I could not find strongly highlited main goal of research which was undertaken by you. I understand the idea of research but the manuscript is lack of main important AIM. I think it would be great if you can describe it in 1-2 sentences on the end of Introduction

-  Materials and Methods have been described in many details (sometimes too many ) and you presented good and well prepared description of obtained results. According to this, I feel that discussion section is too poor in compare to the rest section of manuscript. Based on obtained results you should "rebuilt" the discussion and reached it with more description/comparison with previous results.

- I have no concerns about matherials and methods, which in my opinion are good descibed and prepared. One thing needs to be correct - the title of 5.1 General - it should be changed to Reagents/medium/broths/chemicals ore something in this kind of words.

-The numeration of titles should be checked again, for example Authors used 5. (conclusion and the same for materials and methods; 5.1 for general and the same for Isolation... , 5.5 GC-MS... and the same for NMR, 5.8.3 and others), on this stage of submitting manuscript this kind of mistakes should be out of manuscript

-Similar situation with "scheme" and "figure", it should be one name of this

Author Response

- In my opinion the Introduction section should be shortened and what is more important you should include POISONING aspects of Mercurialis perennis. It is know that this plant can couse hard poisoninf effects, including death. I think that it should be describe and strongly highlithed in Introduction

Response: We have  shorten up the Introduction  and additionally included facts about the poisonous effects of Mercurialis perennis, even the latter aspect was not in the main focus of this manuscript. However, novel investigations on the toxicity of M. perennis are lacking in the literature and reports on fatal (lethal) effects of the herb are ambiguous and probably were due to a confusion with annual dog's mercury (Mercurialis annua). The latter contains 3-cyanopyridine as a poisonous compound, which was not found in M. perennis (for details see our paper:  Lorenz et al.  An approach to the chemotaxonomic differentiation of two European dog’s mercury species: Mercurialis annua L. and M. perennis L. Chem. Biodivers. 2012, 9, 282-294.).

- I could not find strongly highlited main goal of research which was undertaken by you. I understand the idea of research but the manuscript is lack of main important AIM. I think it would be great if you can describe it in 1-2 sentences on the end of Introduction

Response: We redrafted the main aim of our work at the end of the Introduction.

-  Materials and Methods have been described in many details (sometimes too many ) and you presented good and well prepared description of obtained results. According to this, I feel that discussion section is too poor in compare to the rest section of manuscript. Based on obtained results you should "rebuilt" the discussion and reached it with more description/comparison with previous results.

Response: The Discussion section was something extended. More comparisons with previous results have been included.

- I have no concerns about matherials and methods, which in my opinion are good descibed and prepared. One thing needs to be correct - the title of 5.1 General - it should be changed to Reagents/medium/broths/chemicals ore something in this kind of words.

-The numeration of titles should be checked again, for example Authors used 5. (conclusion and the same for materials and methods; 5.1 for general and the same for Isolation... , 5.5 GC-MS... and the same for NMR, 5.8.3 and others), on this stage of submitting manuscript this kind of mistakes should be out of manuscript.

Response: Done! We checked the numbering and re-numbered the respective Sections.

-Similar situation with "scheme" and "figure", it should be one name of this.

Response: Done! We consistently named all Schemes and Figures as "Figures".

Further remarks: We have also deposited now the bacterial DNA sequences in the BLAST database and referred them with the respective accession numbers (see Table 1).